# A Separation Law of Membership Privacy between One- and Two-Layer Networks

## Abstract

We study the problem of identifying whether a target sample is included in the training procedure of neural networks (*i.e.* member *vs.* non-member). This problem is known as the problem of *membership inference attacks*, and raises concerns on the security and privacy of machine learning. In this work, we prove a separation law of membership privacy between one- and two-layer networks: the latter provably preserves less membership privacy against confidence-based attacks than the former. We also prove the phenomenon of *confidence collapse* in two-layer networks, which refers to the phenomenon that the samples of the same class have exactly the same confidence score. Our results are two-fold: a) gradient methods on two-layer ReLU networks converge to a confidence-collapsed solution, such that the attacker can classify members and non-members with perfect precision and recall; b) under the same assumptions as in a), there exists a training dataset such that the confidence collapse phenomenon does not occur and the attacker fails to classify all members and non-members correctly.

## 1 Introduction

A growing body of research shows that neural networks are vulnerable to various threat models, ranging from model inversion (Fredrikson et al., 2015; Wu et al., 2016), model extraction (Tramèr et al., 2016; Wang & Gong, 2018), adversarial examples (Szegedy et al., 2014; Goodfellow et al., 2015) to membership inference (Shokri et al., 2017; Salem et al., 2019), which leads to concerns on the security and privacy of machine learning models. Among the applications, the fields with confidential and sensitive data have received special attention. For example, machine learning models have been developed to predict the health conditions of patients; a successful membership inference attack against a model trained with cancer patients might disclose a victim's confidential health status about cancer, which could cause possible embarrassment and the inability to obtain employment, mortgages, or various forms of insurance. The focus of this paper is on the vulnerability of neural networks to the membership inference attacks. Given a threatened machine learning model and a target sample, the goal of the attacks is to identify whether the target sample was used to train the threatened model.

Membership inference attacks rely on a separation of model behaviors between the training (*a.k.a.* member) and test (*a.k.a.* non-member) data. There are many types of membership inference attacks, such as shadow model based attack (Shokri et al., 2017), confidence score based attack (Salem et al., 2019), and label-only attack (Choquette-Choo et al., 2021). In the shadow model based attack, the ultimate goal of the attacker is to build an attack model to discriminate between the output vectors of the threatened model on the member and non-member data. To achieve this goal, the attacker 1) initializes multiple shadow models of the same architecture, 2) mimics the behavior of the threatened model by training the shadow models on the *i.i.d.* samples of the threatened model, and 3) trains an attack model based on the confidence scores of the shadow models. Later work (Salem et al., 2019) showed that member and non-member data are distinguishable by simply applying a one-sided threshold classifier to the confidence scores of the threatened model.

In this work, we investigate a phenomenon called *confidence collapse* for two-layer networks and the membership privacy. Confidence collapse refers to the phenomenon that the training samples from the same class

have identical confidence score. Once an attacker accesses to the confidence score of one training sample, *e.g.*, by planting a Trojan instance and receiving its confidence score, the attacker is able to determine whether a target sample is a member or non-member via querying its confidence score. We theoretically study the confidence collapse for one- and two-layer neural networks. We prove that, under mild assumptions, confidence collapse always occurs in two-layer ReLU networks trained with gradient descent, but may not occur in their one-layer counterparts trained with the same algorithm. This provides a separation law of membership privacy between one- and two-layer neural networks.

Our study on confidence collapse is motivated by a recently discovered phenomenon of *neural collapse*, which empirically states that the penultimate-layer features of deep neural network trained by cross entropy or mean squared loss concentrate around their class means (Papyan et al., 2020; Han et al., 2021; Zhu et al., 2021; Lu & Steinerberger, 2022). Neural collapse implies confidence collapse, but not vice versa. Although the layer-peeled analysis (Fang et al., 2021; Ji et al., 2021) sheds lights on the neural collapse, the analysis suffers from strong assumptions, *e.g.*, the feature vectors should not depend on the input data. Except the layer-peeled analysis, there are few theoretical results to support the neural collapse phenomenon. In this work, we prove the results of confidence collapse in two-layer ReLU networks without such strong assumptions. The study of confidence collapse might be of independent interest to learning theory more broadly.

## 1.1 Our Main Research Results

Our results consist of two parts: a) two-layer ReLU networks trained by gradient descent provably leak membership privacy (see Section 4); b) one-layer neural networks leak less membership privacy with the same training dataset and training algorithm (see Section 5). Two results are complementary and reveal a separation law of membership privacy between one- and two-layer networks.

**Threat model.** We assume that the attacker can access to the confidence score of the threatened model on one Trojan example $\boldsymbol{x}_0 \in \mathbf{X}$ and a target sample $\boldsymbol{x}$, where $\mathbf{X}$ is the training set. Beyond that, the attacker has no knowledge about the training data or the threatened model. Though a one-sided threshold classifier on the confidence score suffices to build a successful membership inference attack Salem et al. (2019). In this paper we study a more powerful threat model where the attacker can use a two-sided threshold classifier $\mathbb{I}(a \leq |\Phi(\boldsymbol{x}; \boldsymbol{\theta})| \leq b)$ to decide whether a target sample $\boldsymbol{x}$ is a member (if the value is 1) or non-member (if the value is 0), where $\Phi(\boldsymbol{x}; \boldsymbol{\theta})$ is the confidence score in $\mathbb{R}$ for binary classification problems and $\mathbb{I}(\cdot)$ refers the indicator function. Throughout the paper, we fix the threatened model and view $\boldsymbol{x} \sim \mathcal{D}$ as a non-member, where $\mathcal{D}$ is a continuous test distribution. That is, in the evaluation phase we draw a set of samples $\mathbf{X}_{\text{neg}}$ from $\mathcal{D}$ and their ground-truth labels are deemed as non-members.

Before proceeding, we define the *(membership) privacy-preservation region* as follows:

**Definition 1** (privacy-preservation region)**.** *Let $\Phi(\cdot; \boldsymbol{\theta})$ be a neural network which maps an input example to a confidence score in $\mathbb{R}$ and let $\mathbf{X}$ be its training set. We define the privacy-preservation region of $\Phi(\cdot; \boldsymbol{\theta})$ w.r.t. $\mathbf{X}$ by $R(\mathbf{X}, \Phi) := \{\boldsymbol{x} \in \mathbb{R}^d : \gamma \leq |\Phi(\boldsymbol{x}; \boldsymbol{\theta})| \leq \Gamma\}$, where $\gamma = \min_{\boldsymbol{x} \in \mathbf{X}} |\Phi(\boldsymbol{x}; \boldsymbol{\theta})|$ and $\Gamma = \max_{\boldsymbol{x} \in \mathbf{X}} |\Phi(\boldsymbol{x}; \boldsymbol{\theta})|$.*

To justify how $R(\mathbf{X}, \Phi)$ is related to privacy preservation, we highlight that for any non-member $\boldsymbol{x} \sim \mathcal{D}$ in $R(\mathbf{X}, \Phi)$ and any thresholds $a$ and $b$ in the threat model, the attacker cannot identify members $\mathbf{X}$ *vs.* non-member $\boldsymbol{x}$ with 100% precision and recall. This is because in order to correctly recognize all the members, one needs $a \leq \gamma$ and $b \geq \Gamma$. However, the non-member sample $\boldsymbol{x}$ is misclassified as a member since $a \leq |\Phi(\boldsymbol{x}; \boldsymbol{\theta})| \leq b$. We thus use $\text{Pr}_{\mathcal{D}}(R(\mathbf{X}, \Phi))$ to characterize the membership privacy of model $\Phi$ *w.r.t.* a test distribution $\mathcal{D}$. Smaller $\text{Pr}_{\mathcal{D}}(R(\mathbf{X}, \Phi))$ implies that the model is more vulnerable to membership inference attacks, as the confidence score of a target sample $\boldsymbol{x} \sim \mathcal{D}$ is indistinguishable from that of the training data if $\boldsymbol{x}$ lies in the privacy-preservation region. Meanwhile, we also provide a finite-sample proof for the phenomenon of *confidence collapse* for two-layer networks. Confidence collapse provably leaks membership privacy as we show $\text{Pr}_{\mathcal{D}}(R(\mathbf{X}, \Phi)) = 0$. On the other hand, we also provide a counterexample for one-layer networks: confidence collapse does not occur in the same setting, and we show $\text{Pr}_{\mathcal{D}}(R(\mathbf{X}, \Phi)) > 0$.

Below, we state our main theorems informally:

**Theorems 2 and 3** (informal)**.** *Let $\mathcal{D}$ be any continuous input distribution supported on $\mathbb{R}^d$. Suppose that $\Phi$ is trained by gradient descent on the logistic or exponential loss. If the training data size is small enough,*

*we have a) for two-layer ReLU networks, $\mathrm{Pr}_{\mathcal{D}}(R(\mathbf{X}, \Phi)) = 0$ for all training datasets $\mathbf{X}$; b) for one-layer networks, there exists one training dataset $\mathbf{X}$ such that $\mathrm{Pr}_{\mathcal{D}}(R(\mathbf{X}, \Phi)) > 0$.*

Our result highlights a separation law of membership privacy between one- and two-layer networks by comparing the mass of privacy-preservation regions $R(\mathbf{X}, \Phi)$. For two-layer networks where $\mathrm{Pr}_{\mathcal{D}}(R(\mathbf{X}, \Phi)) = 0$, with probability 1 all the vectors in $\mathbf{X}_{\mathrm{neg}}$ do not lie in the privacy-preservation region $R(\mathbf{X}, \Phi)$. The attacker can then query the confidence score of the Trojan example $\boldsymbol{x}_0$ and set $a = b = |\Phi(\boldsymbol{x}_0; \boldsymbol{\theta})|$ to correctly identify all members and non-members. In contrast, for one-layer networks where $\mathrm{Pr}_{\mathcal{D}}(R(\mathbf{X}, \Phi)) > 0$, with probability as high as $(1 - (1 - \mathrm{Pr}_{\mathcal{D}}(R(\mathbf{X}, \Phi)))^{|\mathbf{X}_{\mathrm{neg}}|})$ there exists at least a sample $\boldsymbol{x} \in \mathbf{X}_{\mathrm{neg}}$ such that $\boldsymbol{x} \in R(\mathbf{X}, \Phi)$, which means that the attacker fails to correctly classify all members and non-members and the neural networks leak less membership privacy. Besides, as gradient descent on one-layer networks converges to a hard margin support vector machine (SVM) Soudry et al. (2018), removing any non-support vectors in the training set of the SVM will not change the decision boundary. Thus, information-theoretically one cannot determine whether a non-support vector is a member of one-layer networks or not with *any* threat model including our confidence-based attack.

**Experiments.** We conduct experiments to study the confidence collapse phenomenon on one- and two-layer networks with both synthetic dataset and real world dataset (see Fig. 2). We empirically verify our theoretical result discussed above: confidence collapse occurs in the two-layer ReLU network but not in the one-layer counterpart. Thus, by querying confidence scores, the two-layer ReLU network are easier to leak membership privacy than the one-layer network.

## 1.2 Our Analytical Techniques

We discuss the techniques used for achieving our analytic results.

**Two-layer ReLU networks.** Our results are built upon the phenomenon of confidence collapse of two-layer ReLU neural networks (see Section 4.1). We begin with the result from Lyu & Li (2019), which shows that gradient flow on homogeneous networks converges in direction to the Karush–Kuhn–Tucker (KKT) point of a max-margin problem (see Eq. 4). The stationarity condition of this KKT point induces an explicit formula of the weights w.r.t. KKT multipliers (see Corollary 1) and the complementary slackness shows that if KKT multipliers are greater than zero, the absolute values of confidence scores of corresponding samples must be a fixed constant (see Corollary 2). We then plug the weights into our network, and show by contradiction that when the volume of the training data is small enough, all KKT multipliers are strictly larger than 0 and the confidence collapse occurs. In Section 4.1.1, we remove the small training data assumption by a simple data transformation via padding the input with orthogonal vectors. When the confidence collapse occurs, the privacy-preservation region shrinks to $R(\mathbf{X}, \Phi) = \{\boldsymbol{x} \in \mathbb{R}^d : |\Phi(\boldsymbol{x}; \boldsymbol{\theta})| = \gamma\}$. We show that the probability mass of $R(\mathbf{X}, \Phi)$ is 0 by the continuity of the distribution $\mathcal{D}$ (see Section 4.2). The attacker can then recognize the members and non-members with $100\%$ precision and recall, which yields membership privacy leakage.

**One-layer networks.** We show that there exists a training dataset such that one-layer networks trained on it do not suffer from confidence collapse. The dataset is linearly separable and satisfies all of assumptions that we discussed above. Both one- and two-layer networks are able to reach $100\%$ accuracy on this training set. However, the confidence collapse occurs in two-layer ReLU networks but not the one-layer counterpart. In this case, the privacy-preservation region for the one-layer network is $R(\mathbf{X}, \Phi) = \{\boldsymbol{x} \in \mathbb{R}^d : \gamma \leq |\Phi(\boldsymbol{x}; \boldsymbol{\theta})| \leq \Gamma\}$ for $\Gamma > \gamma$. As $R(\mathbf{X}, \Phi)$ is of non-zero measure on the Euclidean space $\mathbb{R}^d$, the probability mass of $R(\mathbf{X}, \Phi)$ is strictly larger than 0. Thus with high probability, there exists $\boldsymbol{x} \in \mathbf{X}_{\mathrm{neg}}$ such that $\boldsymbol{x} \in R(\mathbf{X}, \Phi)$ and the network preserves at least certain membership privacy.

## 2 Related Works

**Membership inference attack on classification tasks.** In membership inference attack (MIA), an attacker aims to decide whether a sample is used to train a threatened model (Shokri et al., 2017; Salem et al., 2019; Yeom et al., 2018; Shafran et al., 2021; Choquette-Choo et al., 2021; Yu et al., 2021; Rezaei & Liu, 2021; Olatunji et al., 2021; Hui et al., 2021; He et al., 2021; Carlini et al., 2022; Hu et al., 2022). (Shokri et al.,

2017) conducted the first MIA on classification models, which requires multiple shadow models trained on the i.i.d. data. (Salem et al., 2019) argued that the assumptions in shadow training technique are relatively strong, which heavily limit its applications in different scenarios. They proposed a confidence-based attack via focusing on the highest confidence score of the samples. (Yeom et al., 2018) invented the prediction-loss-based MIA, which infers a sample as a member if its prediction loss is smaller than the averaged training loss. This method can be regarded as another types of confidence based attack as the prediction loss is induced by the confidence score. (Choquette-Choo et al., 2021) proposed the label only attack, where the attacker can only access the hard-label output of the target model. They used the robustness of the model against adversarial perturbation to infer membership and achieved comparable performance to the confidence-based attacks. Our work focuses on the confidence based attacks and prove that two-layer ReLU networks leak membership privacy through confidence collapse. Carlini et al. (2022) stated that attacks should be evaluated by computing their true-positive rate at low false-positive rates. In our work, we prove that confidence-based MIA on two-layer network can achieve 100% true-positive rate and 0% false-positive rate. Thus our results still have perfect performance under the metric proposed in Carlini et al.

**Neural collapse.** Neural collapse is a phenomenon in training a neural network, which states that the penultimate-layer features collapse to their class means and the class means centered at their global mean collapse to the vertices of a simplex equiangular tight frame up to scaling. (Papyan et al., 2020) first discovered the neural collapse phenomenon. (Fang et al., 2021; Ji et al., 2021) provided theoretical insight of neural collapse via layer-peeled model with cross-entropy loss. In the layer-peeled model, they assumed the network has infinite representation ability such that the penultimate-layer features are also optimizable. Then the neural network is simplified to a linear network with optimizable inputs. (Han et al., 2021) applied a similar assumption to study the neural collapse with mean-squared loss. However, the layer-peeled assumption is too strong to fit the empirical situations. In our work, we theoretically prove that a weaker version of neural collapse - confidence collapse - occurs in gradient flow of two-layer neural network with only an assumption on the number of training samples.

**Implicit bias in training classifiers.** A line of works Soudry et al. (2018); Ji & Telgarsky (2018); Gunasekar et al. (2018a;b); Nacson et al. (2019); Lyu & Li (2019); Ji & Telgarsky (2020) try to solve the problem that gradient flow/descent is implicitly biased towards solutions with good generalization performance. Gradient flow stands for the gradient descent with infinitesimal step size. (Soudry et al., 2018) showed that full-batch gradient descent on linear logistic regression converges in direction of the max-margin solution of a Support Vector Machine (SVM). (Nacson et al., 2019) analyzed gradient descent for smooth homogeneous models and showed that the parameter converges in direction to a KKT point of the aforementioned max-margin problem. Lyu & Li (2019) weakened the assumptions in (Nacson et al., 2019) for gradient flow on homogeneous models. In our work, we use the KKT conditions of the max-margin problem in Lyu & Li (2019) to prove that gradient flow on a two-layer neural network would lead to confidence collapse.

## 3 Preliminaries

**Notations and problem settings.** We use a *bold capital* letter to represent a matrix or a set of vectors, a *bold lower-case* letter to represent a vector, and a *lower-case* letter to represent a scalar. Specifically, we use $\mathbf{X} := \{\boldsymbol{x}_1, \boldsymbol{x}_2, ..., \boldsymbol{x}_n\} \subseteq \mathbb{R}^d$ to represent the set of training samples with their labels $\{y_i\}_{i=1}^n$ in $\{-1, 1\}$. The probability measure of test distribution $\mathcal{D}$ is denoted by $\Pr_{\mathcal{D}}(\cdot)$. We focus on two-layer ReLU networks for binary classification tasks. We denote the first layer of the ReLU network by $\mathbf{W} := [\boldsymbol{w}_1, ..., \boldsymbol{w}_k] \in \mathbb{R}^{d \times k}$, and the second layer by $\mathbf{v} := (v_1, ..., v_k) \in \mathbb{R}^k$. The ReLU activation function is defined by $\sigma(\cdot) = \max\{\cdot, 0\}$. The confidence (score) of the neural network on a sample $\boldsymbol{x}$ is given by

$$\Phi(\boldsymbol{x}; \boldsymbol{\theta}) = \sum_{i=1}^{k} v_i \sigma(\boldsymbol{w}_i^T \boldsymbol{x}), \tag{1}$$

where $\boldsymbol{\theta}$ is the concatenation of $\mathbf{W}$ and $\mathbf{v}$. A neural network is called *homogeneous* if there is a constant $L \geq 0$, such that $\forall z > 0, \ \Phi(\boldsymbol{x}; z\boldsymbol{\theta}) = z^L \Phi(\boldsymbol{x}; \boldsymbol{\theta})$. The two-layer ReLU neural network studied in this work is homogeneous with $L = 2$. We will use $\|\cdot\|_2$ to represent the $\ell_2$-norm of a vector.

**Evaluation of MIA.** Following the previous works Shokri et al. (2017); Salem et al. (2019), we use a mixture of training data $\mathbf{X}$ and randomly generated data $\mathbf{X}_{\text{neg}}$ from $\mathcal{D}$ to evaluate the attacker. Note that with probability 1, $\mathbf{X} \cap \mathbf{X}_{\text{neg}} = \emptyset$, as $\mathcal{D}$ is a continuous distribution. We apply precision-recall to measure the quality of the attacker. Denote by $\mathbf{A}_{\text{pos}}$ the set of attacker's selected members, and $\mathbf{A}_{\text{neg}}$ the non-members. The precision and recall are calculated by $|\mathbf{A}_{\text{pos}} \cap \mathbf{X}|/|\mathbf{A}_{\text{pos}}|$ and $|\mathbf{A}_{\text{pos}} \cap \mathbf{X}|/|\mathbf{X}|$, respectively. Higher scores of precision and recall imply more severe leakage of membership privacy. We say *perfect privacy leakage* occurs if and only if both precision and recall are 1.

**Confidence collapse.** Below, we define confidence collapse for binary classification tasks:

**Definition 2** (Confidence collapse)**.** *In binary classification tasks, given a training set $\mathbf{X}$, we say that confidence collapse occurs in a neural network $\Phi$ parameterized by $\boldsymbol{\theta}$ if and only if*

$$\exists C > 0, \ \forall \boldsymbol{x} \in \mathbf{X}, \ |\Phi(\boldsymbol{x}; \boldsymbol{\theta})| = C.$$

For a homogeneous neural network, the confidence collapse phenomenon is scaling-invariant *w.r.t.* the weights, because if $|\Phi(\boldsymbol{x}; \boldsymbol{\theta})| = C$, then $\forall \alpha > 0, |\Phi(\boldsymbol{x}; \alpha\boldsymbol{\theta})| = \alpha^L C$, which indicates that neural networks with weights $\alpha\boldsymbol{\theta}, \alpha > 0$ also suffer from confidence collapse. Thus, confidence collapse of networks depends only on the direction of $\boldsymbol{\theta}$, *i.e.*, $\frac{\boldsymbol{\theta}}{||\boldsymbol{\theta}||_2}$, rather than the scaling of $\boldsymbol{\theta}$.

**Gradient flow on homogeneous neural networks.** For a training set $\{(\boldsymbol{x}_i, y_i)\}_{i=1}^n \subseteq \mathbb{R}^d \times \{-1, 1\}$, let $\Phi(\cdot; \boldsymbol{\theta}) : \mathbb{R}^d \to \mathbb{R}$ be a neural network mapping with weights $\boldsymbol{\theta}$. The empirical loss *w.r.t.* $\Phi(\cdot; \boldsymbol{\theta})$ and the loss function $l(\cdot) : \mathbb{R} \to \mathbb{R}$ is:

$$L(\boldsymbol{\theta}) := \sum_{i=1}^n l(y_i \Phi(\boldsymbol{x}_i; \boldsymbol{\theta})). \tag{2}$$

In this work, we focus on the exponential loss $l(t) = e^{-t}$ and logistic loss $l(t) = \log(1 + e^{-t})$. We analyze the optimization problem of Eq. 2 with gradient flow—gradient descent with infinitesimal step size. In this setting, the learnable parameter $\boldsymbol{\theta}$ changes continuously with time $t$. We define the trajectory of $\boldsymbol{\theta}$ by $\boldsymbol{\theta}(t)$. If $\lim_{t \to \infty} \frac{\boldsymbol{\theta}(t)}{||\boldsymbol{\theta}(t)||_2} = \frac{\hat{\boldsymbol{\theta}}}{||\hat{\boldsymbol{\theta}}||_2}$, we say the trajectory of $\boldsymbol{\theta}$ converges *in direction* to $\hat{\boldsymbol{\theta}}$. We use the result in Lyu & Li (2019), which shows that gradient flow on a homogeneous neural network converges *in direction* to a KKT point of the following max-margin problem,

$$\min_{\boldsymbol{\theta}} \frac{1}{2} ||\boldsymbol{\theta}||_2^2, \ \text{s.t.} \ y_i \Phi(\boldsymbol{x}_i; \boldsymbol{\theta}) \geq 1, \quad \forall i \in [n], \tag{3}$$

if there exists time $t_0$, such that $\Phi(\cdot; \boldsymbol{\theta}(t_0))$ has 100% accuracy on training set. We defer the details of their theorem to Lemma 1 in the Appendix.

## 4 Two-Layer Networks Provably Leak Membership Privacy

In this section, we present our theoretical contributions and show that two-layer ReLU networks provably leak membership privacy. Our proof consists of two aspects: 1) confidence collapse in two-layer ReLU networks is an inevitable consequence of training by gradient methods; 2) given a continuous data distribution $\mathcal{D}$, the probabilistic mass of associated privacy-preservation region is 0. With 1) and 2), a confidence-based attacker can determine whether a given example is in the training set or not by its confidence score.

### 4.1 Confidence Collapse on Two-Layer Networks

Our analysis demonstrates a confidence collapse phenomenon of two-layer ReLU networks that are optimized by gradient descent. Based on the result of Lyu & Li (2019), gradient flow on the two-layer ReLU network $\Phi$ with weight $\boldsymbol{\theta} = [\mathbf{W}, \mathbf{v}]$ converges in direction to the KKT point of a maximum margin problem Eq. 3. As discussed previously, the confidence collapse phenomenon is scaling-invariant w.r.t. the weights on homogeneous networks, we only need to show that confidence collapse occurs when it comes to the KKT solution of the above maximum margin problem. For simplification, we denote $\Phi(\cdot; \boldsymbol{\theta})$ by $\Phi(\cdot)$.

**Theorem 1.** *Given $n$ samples $\{(\boldsymbol{x}_i, y_i)\}_{i=1}^n$ in $\mathbb{R}^d \times \{-1, 1\}$, consider a two-layer ReLU neural network $\Phi(\cdot)$ with $k$ hidden neurons on the binary classification problem. Let $[\mathbf{W}, \mathbf{v}]$ be a KKT point of the max-margin problem (Eq. 3). Denote by $p := \max_{i,j \in [n], \boldsymbol{x}_j \neq \pm \boldsymbol{x}_i} |\boldsymbol{x}_i^T \boldsymbol{x}_j|$ and $q := \min_{i \in [n]} \|\boldsymbol{x}_i\|_2^2$. If $n < \min\{d, \frac{q}{3p}\}$, we have*

$$\forall i \in [n], \quad y_i \Phi(\boldsymbol{x}_i) = 1.$$

Theorem 1 states that, a two-layer ReLU network trained with finite data converges in direction to a solution where the confidence scores of all training samples are the same. Under the setting of homogeneous networks, Theorem 1 implies the confidence collapse of two-layer ReLU network, but probably with a different confidence score $C$. For example, if all training samples are orthogonal in $\mathbb{R}^d$, then $\min\{d, \frac{q}{3p}\} = d$. By Theorem 1, when $n \leq d$, the confidence scores of the $n$ orthogonal examples collapse. Our theoretical analysis does not imply that the network converges to 0 loss. In practice, when the dataset size is large, confidence collapse will not occur on the two-layer network and we cannot use confidence-based attacks to achieve perfect MIA on two-layer networks. Duplication of images doesn't affect confidence collapse in practice. This observation indicates that adding more data is a good way to defend the confidence-based attack we propose.

Despite our strong conclusion on confidence collapse, the assumptions in our theorem are moderate. Our theorem does not make any assumptions on the scale of the input data. The only limitation of our theorem is the assumption of small training data: the number of training data should be fewer than $\min\{d, \frac{q}{3p}\}$. In the following subsection, we show how we can remove this assumption by simple data transformation.

### 4.1.1 Simple data transformation removes the small data assumption

We will show how simple data transformation can help us remove the small data assumption in Theorem 1. Our technique is to pad the input with orthogonal vectors.

**Proposition 1.** *Given $n$ samples $\{(\boldsymbol{x}_i, y_i)\}_{i=1}^n$ in $\mathbb{R}^d \times \{-1, 1\}$, let $p' := \max_{i,j \in [n]} |\boldsymbol{x}_i^T \boldsymbol{x}_j|$. Consider a set of mutually orthogonal vectors $\{\mathbf{a}_i \in \mathbb{R}^n\}_{i=1}^n$ with $\|\mathbf{a}_i\|_2 = \sqrt{3np'}$, and a two-layer ReLU neural network $\Phi(\cdot)$ with $k$ hidden neurons on the new training dataset $\{([\boldsymbol{x}_i, \mathbf{a}_i], y_i)\}_{i=1}^n$. Let $[\mathbf{W}, \mathbf{v}]$ be a KKT point of the max-margin problem (Eq. 3). We have*

$$\forall i \in [n], \quad y_i \Phi(\boldsymbol{x}_i) = 1.$$

Proposition 1 states that, if the training samples are padded with a set of mutually orthogonal vectors, confidence collapse occurs no matter how large the training dataset is. For the padding vector $\{\mathbf{a}_i\}_{i=1}^n \subseteq \mathbb{R}^n$, we can simply set them as the standard basis of $\mathbb{R}^n$ scaled by $\sqrt{3np'}$, *i.e.*, $\mathbf{a}_i = (0, 0, \cdots, \sqrt{3np'}, \cdots, 0)$, where only the $i$-th entry is $\sqrt{3np'}$. Intuitively, this data transformation assigns each sample with its identity pseudo-label, and uses the one-hot encoding of the pseudo-label as additional features during training. The pseudo-labels provides additional information about the identity of each sample and thus will not conflict with the 100% accuracy assumption of the classifier.

### 4.1.2 Proof sketch of Theorem 1 and Proposition 1

**Proof sketch of Theorem 1.** Denote by $J := [k]$, $J_+ = \{i \in [k] : v_i \geq 0\}$, $J_- = \{i \in [k] : v_i < 0\}$ and $I := [n]$, $I_+ = \{i \in [n] : y_i = -1\}$, $I_- = \{i \in [N] : y_i = 1\}$. Based on the result of (Lyu & Li, 2019) and Eq. 3, gradient flow on the two-layer ReLU network $\Phi$ with weight $\mathbf{W}$ and $\mathbf{v}$ converges in direction to the KKT point of a maximum margin problem

$$\min_{\boldsymbol{\theta}} \frac{1}{2} \left( \|\mathbf{v}\|_2^2 + \sum_{i=1}^k \|\boldsymbol{w}_i\|_2^2 \right), \text{ s.t. } y_i \Phi(\boldsymbol{x}_i) = y_i \sum_{i=1}^k v_i \sigma(\boldsymbol{w}_i^T \boldsymbol{x}_i) \geq 1, \; \forall i \in [n], \tag{4}$$

where $k$ is the width of the network. By applying the KKT stationarity and complementary slackness conditions, the optimization problem Eq. 4 naturally implies the following properties of two-layer ReLU networks.

**Corollary 1.** *For two-layer ReLU networks (Eq. 1), by the stationarity condition in KKT, there exists $\lambda_j \geq 0, j \in [n]$ such that for all $s \in [k]$,*

$$\boldsymbol{w}_s = \sum_{i \in I} \lambda_i \nabla_{\boldsymbol{w}_s}(y_i \Phi(\boldsymbol{x}_i)) = \sum_{i \in I} \lambda_i y_i v_s \sigma'_{i,s} \boldsymbol{x}_i, \tag{5}$$

*where $\sigma'_{i,s} := \mathbb{I}(\boldsymbol{w}_s^T \boldsymbol{x}_i > 0)$.*

**Corollary 2.** *For two-layer ReLU networks (Eq. 1), by the complementary slackness condition in KKT, we have $\lambda_i(y_i \Phi(\boldsymbol{x}_i) - 1) = 0, \ \forall i \in [n]$.*

Corollary 1 identify the form of the first-layer weights. We plug this form of weights into the network $\Phi(\boldsymbol{x})$ and obtain

$$\Phi(\boldsymbol{x}) = \sum_{j \in J} v_j \sigma(\sum_{i \in I} \lambda_i y_i v_j \sigma'_{i,j} \boldsymbol{x}_i^T \boldsymbol{x}).$$

Corollary 2 shows the relation between the KKT multipliers and the associated constrains. In our proof, we use the properties that $y_i \Phi(\boldsymbol{x}_i) > 1 \Rightarrow \lambda_i = 0$ and $\lambda_i > 0 \Rightarrow y_i \Phi(\boldsymbol{x}_i) = 1$.

Now we are ready to prove Theorem 1 by contradiction. Assume that there exists $l \in [n]$ such that $y_l \Phi(\boldsymbol{x}_l) > 1$. Without loss of generality, let $l \in I_+$. The proof for the case of $l \in I_-$ is similar. Our proof consists of four steps:

**Step 1.** We show $\max_{i \in I} \sum_{j \in J_+} \lambda_i v_j^2 \sigma'_{i,j} > \frac{1}{np}$. By the complementary slackness condition (Corollary 2) of the KKT point, $\lambda_l(y_l \Phi(\boldsymbol{x}_l) - 1) = 0$. As $y_l \Phi(\boldsymbol{x}_l) - 1 > 0$, we have $\lambda_l = 0$. Then, by the stationarity condition of the KKT point (Corollary 1), we have $\boldsymbol{w}_s = \sum_{i \in I} \lambda_i y_i v_s \sigma'_{i,s} \boldsymbol{x}_i$. We can then plug it into $1 < y_l \Phi(\boldsymbol{x}_l) = \sum_{i=1}^{k} v_i \sigma(\boldsymbol{w}_i^T \boldsymbol{x}_l)$, which induces $\max_{i \in I} \sum_{j \in J_+} \lambda_i v_j^2 \sigma'_{i,j} > \frac{1}{np}$.

**Step 2.** We show that $y_r \Phi(\boldsymbol{x}_r) = 1$ for $r := \arg\max_{i \in I} \sum_{j \in J_+} \lambda_i v_j^2 \sigma'_{i,j}$. We assume without loss of generality that $\max_{i \in I} \sum_{j \in J_+} \lambda_i v_j^2 \sigma'_{i,j} \geq \max_{i \in I} \sum_{j \in J_-} \lambda_i v_j^2 \sigma'_{i,j}$. We have $\lambda_r > 0$ because $\lambda_r \sum_{j \in J_+} v_j^2 \sigma'_{r,j} > \frac{1}{np} > 0$. Thus, by Corollary 2, we have $y_r \Phi(\boldsymbol{x}_r) = 1$.

**Step 3.** We derive a contradiction for $r \in I_-$. We first plug $\boldsymbol{w}_s = \sum_{i \in I} \lambda_i y_i v_s \sigma'_{i,s} \boldsymbol{x}_i$ into $y_r \Phi(\boldsymbol{x}_r)$. If $r \in I_-$, we have $\lambda_r v_j \sigma'_{r,j} q \leq \sum_{i \in I} \lambda_i v_j \sigma'_{i,j} p$ for $j \in J_+$. Summing it over $j \in J_+$ yields $\sum_{j \in J_+} \lambda_r v_j^2 \sigma'_{r,j} \leq \frac{1}{3} \max_{i \in I} \sum_{j \in J_+} \lambda_i v_j^2 \sigma'_{i,j}$, which contradicts to $r = \arg\max_{i \in I} \sum_{j \in J_+} \lambda_i v_j^2 \sigma'_{i,j}$.

**Step 4.** We derive a contradiction for $r \in I_+$. If $r \in I_+$, by plugging $\boldsymbol{w}_s = \sum_{i \in I} \lambda_i y_i v_s \sigma'_{i,s} \boldsymbol{x}_i$ into $y_r \Phi(\boldsymbol{x}_r)$, we have $1 = \sum_{j=1}^{k} v_j \sigma(\boldsymbol{w}_j^T \boldsymbol{x}_r) \geq (q - 2np) \max_{i \in I} \sum_{j \in J_+} \lambda_i v_j^2 \sigma'_{i,j} > \frac{(q-2np)}{np}$, which contradicts to the setting that $n < q/3p$.

Following Steps 1-4, we have $\forall i \in [n], y_i \Phi(\boldsymbol{x}_i) = 1$. The complete proof is in Appendix A.

**Proof sketch of Proposition 1.** Consider the new training set $\{(\boldsymbol{x}_i' := [\boldsymbol{x}_i, \mathbf{a}_i], y_i)\}_{i=1}^{n}$. Firstly, the norm of the new sample $\boldsymbol{x}_i'$ is given by $\sqrt{||\boldsymbol{x}_i||_2^2 + ||\mathbf{a}_i||_2^2} = \sqrt{||\boldsymbol{x}_i||_2^2 + 3np'}$. The inner product of $\boldsymbol{x}_i'$ and $\boldsymbol{x}_j'$ $(i \neq j)$ is $\boldsymbol{x}_i'^T \boldsymbol{x}_j' = \boldsymbol{x}_i^T \boldsymbol{x}_j + \mathbf{a}_i^T \mathbf{a}_j = \boldsymbol{x}_i^T \boldsymbol{x}_j$. Thus

$$p := \max_{i,j \in [n], \boldsymbol{x}_j \neq \pm \boldsymbol{x}_i} |\boldsymbol{x}_i'^T \boldsymbol{x}_j'| \leq \max_{i,j \in I} |\boldsymbol{x}_i^T \boldsymbol{x}_j| =: p'.$$

As the new dataset $\{(\boldsymbol{x}_i', y_i)\}_{i=1}^{n}$ is in $\mathbb{R}^{n+d} \times \{-1, 1\}$, we have a) $n + d \geq n$, and b) $\frac{\min_{i \in [n]} ||\boldsymbol{x}_i'||_2^2}{3p} \geq \frac{3np'}{3p} \geq \frac{3np}{3p} = n$. Thus $n \leq \min\{n + d, \frac{\min_{i \in [n]} ||\boldsymbol{x}_i'||_2^2}{3p'}\}$ always holds. By applying Theorem 1 to the new dataset $\{(\boldsymbol{x}_i', y_i)\}_{i=1}^{n}$, we have that the confidence collapse occurs on $\{(\boldsymbol{x}_i', y_i)\}_{i=1}^{n}$. The complete proofs are in Appendix B.

## 4.2 Vulnerability of Two-Layer Networks to Membership Inference Attacks

We now prove that when confidence collapse occurs, the probabilistic mass of the privacy-preservation region w.r.t. $\mathcal{D}$ is 0. Thus, if the test examples are generated from $\mathcal{D}$, their confidence should be different from the

confidence score of the training data almost surely. By Definition 1 and Definition 2, the privacy-preservation region with confidence collapse (score $\gamma$) is given by $R(\mathbf{X}, \Phi) = \{\boldsymbol{x} \in \mathbb{R}^d : \Phi(\boldsymbol{x}) = \pm\gamma\}$.

**Theorem 2.** *Given $R(\mathbf{X}, \Phi) = \{\boldsymbol{x} \in \mathbb{R}^d : \Phi(\boldsymbol{x}) = \pm\gamma\}$, where $\Phi(\cdot)$ is an arbitrary two-layer ReLU network and $\mathcal{D}$ is a continuous distribution in $\mathbb{R}^d$, we have $\Pr_{\mathcal{D}}(R(\mathbf{X}, \Phi)) = 0$.*

Theorem 2 states that, if the confidence collapse occurs on the two-layer ReLU network, its privacy-preservation region is of zero measure w.r.t. distribution $\mathcal{D}$. In the test phase, we use a mixture of training data $\mathbf{X}$ and randomly generated data $\mathbf{X}_{\text{neg}}$ from $\mathcal{D}$ to evaluate the attacker. With $\Pr_{\mathcal{D}}(R(\mathbf{X}, \Phi)) = 0$, the absolute value of the confidence score of a randomly generated data almost surely differs from $\gamma$. Thus the attacker can use the confidence score of the Trojan sample to determine the unknown $\gamma$. Then by setting $a = b = \gamma$ in $\mathbb{I}(a \leq |\Phi(\cdot)| \leq b)$, the attacker can apply $\mathbb{I}(|\Phi(\cdot)| = \gamma)$ to identify members $\mathbf{X}$ and non-members $\mathbf{X}_{\text{neg}}$ with 100% precision and recall. In conclusion, confidence collapse almost surely leaks membership privacy. Combining Theorem 1 and Theorem 2, we conclude that when the number of training data is limited, two-layer ReLU networks provably leak membership privacy under our threat model.

**Proof sketch.** We calculate the probabilistic mass of $\{\boldsymbol{x} \in \mathbb{R}^d : \Phi(\boldsymbol{x}) = \pm\gamma\}$ under the distribution $\mathcal{D}$. Firstly, we consider the solution of $\Phi(\boldsymbol{x}) = \pm\gamma$. By the stationarity condition of the KKT point (Corollary 1), we have $\boldsymbol{w}_s = \sum_{i \in I} \lambda_i y_i v_s \sigma'_{i,s} \boldsymbol{x}_i$. Plugging it into $\Phi(\boldsymbol{x}) = \pm\gamma$, where $\boldsymbol{x} = (x^{(1)}, ..., x^{(d)})$, we obtain

$$\Phi(\boldsymbol{x}) = \sum_{j \in J} v_j \mathbb{I}(\{\boldsymbol{w}_j^T \boldsymbol{x} > 0\}) \sum_{i=1}^{d} w_{ji} x^{(i)} = \sum_{i=1}^{d} \left( \sum_{j \in J} v_j \mathbb{I}(\{\boldsymbol{w}_j^T \boldsymbol{x} > 0\}) w_{ji} \right) x^{(i)} = \pm\gamma.$$

Let $a_j(\boldsymbol{x}) = \sum_{j \in J} v_j \mathbb{I}(\{\boldsymbol{w}_j^T \boldsymbol{x} > 0\}) w_{ji}$. Note that $a_j(\boldsymbol{x})$ can take at most $2^j$ values for different $\boldsymbol{x}$. Thus $\mathbf{a}(\boldsymbol{x}) = (a_1(\boldsymbol{x}), ..., a_d(\boldsymbol{x}))$ has at most $2^{jd}$ different choices. Denote by $A := \{\mathbf{a}(\boldsymbol{x}) : \boldsymbol{x} \in \mathbb{R}^d\}$ and $S(\mathbf{a}) := \{\boldsymbol{x} \in \mathbb{R}^d : \mathbf{a}^T \boldsymbol{x} = \pm\gamma\}$, we have $R(\mathbf{X}, \Phi) \subseteq \cup_{\mathbf{a} \in A} S(\mathbf{a})$. Notice that $S(\mathbf{a})$ consists of points on two hyperplanes $\mathbf{a}^T \boldsymbol{x} = \pm\gamma$. For an arbitrary continuous distribution on $\mathbb{R}^d$, we have $\Pr_{\mathcal{D}}(S(\mathbf{a})) = 0$. Thus

$$\Pr_{\mathcal{D}}(R(\mathbf{X}, \Phi)) \leq \sum_{\mathbf{a} \in A} \Pr_{\mathcal{D}}(S(\mathbf{a})) = |A| \cdot 0 = 0.$$

That is, $\Pr_{\boldsymbol{x}_{test} \sim \mathcal{D}}(\boldsymbol{x}_{test} \in R(\mathbf{X}, \Phi)) = 0$. The complete proof is in Appendix C.

One potential way for extending our results to multi-class classification is to convert the problem to the binary case with one vs rest loss, i.e., we define the confidence score of multi-class classification as $f_y(x) - \max_{i \in [k], i \neq y} f_i(x)$ for a given sample $(x, y)$ and classifier $f$. We will leave the detail analysis as a future work.

## 5 One-Layer Networks Leak Less Membership Privacy

In this section, we provide an example where one-layer networks do not suffer from confidence collapse under the same assumptions. Denote the network by $\Phi(\boldsymbol{x}) = \boldsymbol{w}^T \boldsymbol{x} + b$. The existing works (Lyu & Li, 2019; Soudry et al., 2018) show that the gradient flow on one-layer networks with linearly separable datasets converges in direction to a max-margin solution of Eq. 3 with $\theta = [\boldsymbol{w}, b]$, which is a hard-margin SVM. As discussed previously, the confidence collapse phenomenon is scaling-invariant w.r.t. the weights on homogeneous networks, we only need to show the confidence collapse does not occur in the KKT point of the above maximum margin problem. With this, we prove that the probabilistic mass of the corresponding privacy-preservation region $R(\mathbf{X}, \Phi)$ with continuous $\mathcal{D}$ supported on $\mathbb{R}^d$ is strictly larger than 0. Recall that according to Definition 1, $R(\mathbf{X}, \Phi) := \{\boldsymbol{x} \in \mathbb{R}^d : \gamma \leq |\boldsymbol{w}^T \boldsymbol{x} + b| \leq \Gamma\}$, where $\gamma = \min_{\boldsymbol{x} \in \mathbf{X}} |\Phi(\boldsymbol{x})|$ and $\Gamma = \max_{\boldsymbol{x} \in \mathbf{X}} |\Phi(\boldsymbol{x})|$.

**Theorem 3.** *There exists a linearly separable training set $\{(\boldsymbol{x}_i, y_i)\}_{i=1}^{n}$ in $\mathbb{R}^d \times \{-1, 1\}$, which satisfies all the assumptions on training data in Theorem 1, i.e., $n < \min\{d, \frac{q}{3p}\}$, $p := \max_{i,j \in [n], \boldsymbol{x}_j \neq \pm \boldsymbol{x}_i} |\boldsymbol{x}_i^T \boldsymbol{x}_j|$, and $q := \min_{i \in [n]} ||\boldsymbol{x}_i||_2$, such that the KKT point of Eq. 6 is not a confidence-collapsed solution, and $\Pr_{\mathcal{D}}(R(\mathbf{X}, \Phi)) > 0$.*

Theorem 3 states that, under the same data assumptions as Theorem 1, there exists at least one dataset, such that confidence collapse does not hold true for one-layer networks. We also show that the probabilistic

mass of its privacy-preservation region is non-zero. In the test phase, with probability as high as $(1 - (1 - \text{Pr}_{\mathcal{D}}(R(\mathbf{X}, \Phi)))^{|\mathbf{X}_{\text{neg}}|})$, there exists at least one sample $\boldsymbol{x} \in \mathbf{X}_{\text{neg}}$ such that $\boldsymbol{x} \in R(\mathbf{X}, \Phi)$. With this, the attacker fails to identify members $\mathbf{X}$ *vs.* non-member $\boldsymbol{x}$ with 100% precision and recall. The reason is that to correctly recognize all members (100% precision), we need $a \leq \gamma$ and $b \geq \Gamma$ for the indicator $\mathbb{I}(a \leq |\Phi(\cdot)| \leq b)$. However, the sample $\boldsymbol{x}$ is misclassified as a member since $a \leq |\Phi(\boldsymbol{x})| \leq b$. Compared to two-layer networks, one-layer networks leak less membership privacy under our threat model. Besides, removing any non-support vectors in the training set of the hard-margin SVM will not change the decision boundary. Therefore, information-theoretically one cannot determine whether a non-support vector is a member or not with *any* threat model including our confidence-based attack.

**Proof sketch.** We prove Theorem 3 by constructing a hard instance, which satisfies the assumption on the training dataset but does not suffer from confidence collapse by one-layer neural networks. Based on Eq. 3, the max-margin problem is formulated by

$$\min_{\boldsymbol{\theta}} \frac{1}{2}(\|\boldsymbol{w}\|_2^2 + b^2), \text{ s.t. } y_i(\boldsymbol{w}^T \boldsymbol{x}_i + b) \geq 1, \quad \forall i \in [n]. \tag{6}$$

Consider $(\boldsymbol{x}_1, +1), (\boldsymbol{x}_2, +1), (\boldsymbol{x}_3, +1), (\boldsymbol{x}_4, -1)$, where $\boldsymbol{x}_i \in \mathbb{R}^d, d > 4, \|\boldsymbol{x}_i\|_2 = \sqrt{d}, \boldsymbol{x}_2 = -\boldsymbol{x}_1, \boldsymbol{x}_3^T \boldsymbol{x}_1 = 0, \boldsymbol{x}_4^T \boldsymbol{x}_1 = 0, \boldsymbol{x}_3^T \boldsymbol{x}_4 = -\epsilon$, and $\epsilon$ is an arbitrary positive constant less than 1/6 (*e.g.*, $\boldsymbol{x}_1 = (\sqrt{d}, 0, \cdots, 0)$, $\boldsymbol{x}_2 = (-\sqrt{d}, 0, \cdots, 0)$, $\boldsymbol{x}_3 = (0, \sqrt{d}, 0, \cdots, 0)$, and $\boldsymbol{x}_4 = (0, \frac{-\epsilon}{\sqrt{d}}, \sqrt{d - \frac{\epsilon^2}{d}}, 0, \cdots, 0))$.

Firstly, let $q := \min_{i \in [n]} \|\boldsymbol{x}_i\|_2 = \sqrt{d}$ and $p := \max_{i,j \in [n], \boldsymbol{x}_j \neq \pm \boldsymbol{x}_i} |\boldsymbol{x}_i^T \boldsymbol{x}_j| = \epsilon$. We can easily verify that $4 < \min\{d, \frac{q}{3p}\}$. Besides, we can use a one-layer neural network $\Phi(\boldsymbol{x}) = (\boldsymbol{x}_3 - \boldsymbol{x}_4)^T \boldsymbol{x} + 1$ to classify them correctly, which means the dataset is linearly separable. Next, following the KKT stationarity condition of the optimization problem Eq. 6, there exist $\lambda_i \geq 0, i \in [4]$ such that $\boldsymbol{w} = \sum_{i=1}^4 \lambda_i y_i \boldsymbol{x}_i$. For the optimization problem Eq. 6, confidence collapse only occurs with $y_i \Phi(\boldsymbol{x}_i) = 1, i \in [4]$. Otherwise by complementary slackness we will have $\lambda_i = 0, i \in [4]$, which implies $\boldsymbol{w} = b = 0$ and all samples are misclassified. Pluging $\boldsymbol{w}$ into $y_i \Phi(\boldsymbol{x}_i) = 1, i \in [4]$ will reach a contradiction. Thus, one-layer networks do not converge to a model with confidence collapse under the selected training samples.

Denote by $\gamma = \arg\min_{\boldsymbol{x} \in \mathbf{X}} |\boldsymbol{w}^T \boldsymbol{x} + b|$ and $\Gamma = \arg\max_{\boldsymbol{x} \in \mathbf{X}} |\boldsymbol{w}^T \boldsymbol{x} + b|$. Confidence non-collapse implies $\Gamma$ is strictly larger than $\gamma$. By Definition 1, we have $R(\mathbf{X}, \Phi) = \{\boldsymbol{x} \in \mathbb{R}^d : \gamma \leq |\boldsymbol{w}^T \boldsymbol{x} + b| \leq \Gamma\}$. We can rotate the coordinate such that $\boldsymbol{w}^T = (\|\boldsymbol{w}\|_2, 0, ..., 0)$. Let $\boldsymbol{x} = (x^{(1)}, ..., x^{(d)})$ and $\mathcal{D}'$ be the new distribution after rotation. We derive $\text{Pr}_{\mathcal{D}}(R(\mathbf{X}, \Phi)) > 0$ because $\mathcal{D}$ is supported on $\mathbb{R}^d$. The complete proof is in Appendix D.

# 6 Experiments

We conduct experiments on synthetic data, MNIST, and CIFAR10 to validate the confidence collapse phenomenon. Our experiments are run on a 24GB Nvidia A5000 GPU. We train model with cross-entropy loss and report the logits as the confidence score. One can normalize the logit confidence to [0,1] by applying the Sigmoid activation.

## 6.1 Synthetic data

**Settings.** We conduct a synthetic experiment to verify the confidence collapse phenomenon on one- and two-layer networks (see Fig. 2). Our experiments are run on a 24GB Nvidia Tesla P40 GPU. We randomly sample 10,000 examples from Gaussian distribution on $\mathbb{R}^{20,000}$ as the training dataset, which satisfies the data assumptions in Theorem 1. For the two-layer network, we set the width of the hidden layer as 1,000. The number of training epochs is 40,000 and the learning rate is fixed as 0.0008 throughout the training procedure for both networks.

**Results.** From Fig. 2 top and bottom left, we can see that confidence collapse occurs in the two-layer ReLU network but not in the one-layer counterpart. Clearly, by querying confidence scores, the two-layer ReLU network are easier to leak membership privacy than the one-layer network.

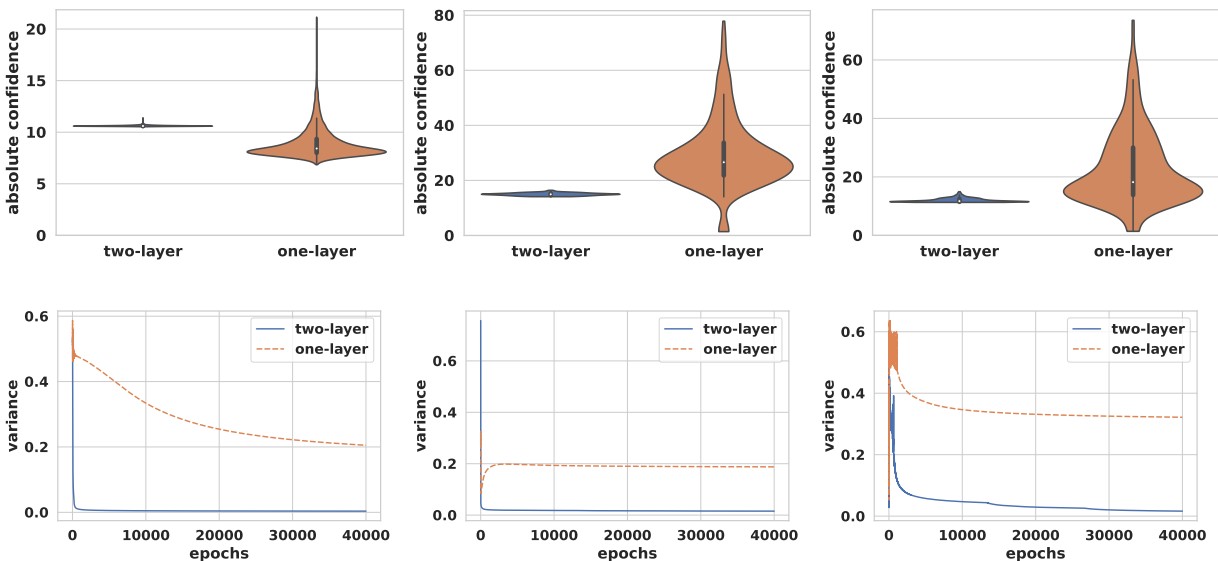

Figure 1: **Top:** violin plot of absolute values of confidence scores on the synthetic (**left**), MNIST (**middle**), and CIFAR10 (**right**) training examples by one- and two-layer networks. **Bottom:** variance of absolute values of confidence scores on the synthetic (**left**), MNIST (**middle**), and CIFAR10 (**right**) training examples across different epochs. It shows that confidence collapse occurs in the two-layer ReLU network but not in the one-layer counterpart. From the confidence scores, it is easy to infer whether a given sample is involved in the training phase.

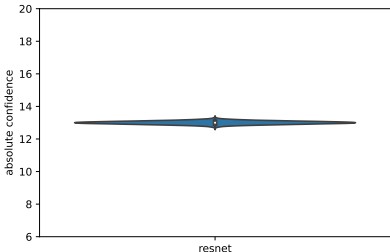

Figure 2: Violin plot of confidence scores on CIFAR10 with ResNet18.

## 6.2 MNIST and CIFAR10

**Settings.** As our paper focus on training-time binary classification problem, we select two classes from MNIST and CIFAR10 for training and neglect the test set of our benchmarks. For the two-layer network on MNIST, the first layer is of size (28×28, 500) and the second layer is of size (500, 1). We use ReLU activation between the two layers and apply SGD with learning rate 0.001 to optimize it. For the one-layer network on MNIST, the layer is of size (28×28, 1) and we apply SGD with learning rate 0.001 to optimize it. For the two-layer network on CIFAR10, the first layer is of size (3×32×32, 1000) and the second layer is of size (1000, 1). We use ReLU activation between the two layers and apply SGD with learning rate 0.0001 to optimize it. For the one-layer network on CIFAR10, the layer is of size (3×32×32, 1) and we apply SGD with learning rate 0.00015 to optimize it.

**Results.** Fig. 2 top and bottom (middle) figures illustrate the absolute value of confidence score and its variance (during training) for MNIST. Fig. 2 top and bottom (right) figures illustrate the absolute value of confidence score and its variance (during training) for CIFAR10. All experiments are run over 40,000 epochs, and we report the confidence score related measurements over all epochs. From these plots we can

see that confidence collapse occurs in the two-layer ReLU network but not in the one-layer counterpart for MNIST and CIFAR10. Thus, by querying confidence scores, the two-layer ReLU network are easier to leak membership privacy than the one-layer network.

## 7 Discussions and Conclusion

**Limitation.** We assume that our networks are optimized by gradient flow, which requires the step size of gradient descent to be arbitrarily small. However, we point out that gradient flow is a frequently used tool to analyze the dynamics of neural networks. For example, except its applications to implicit bias Lyu & Li (2019); Soudry et al. (2018), gradient flow also serves as a key assumption in neural tangent kernel Jacot et al. (2018), overparameterized feature learning Chen et al. (2022), robustness of neural networks Vardi et al. (2022), *etc.* Lyu & Li (2019) also showed that gradient descent on homogeneous networks converges approximately to the KKT point of the max-margin problem Eq. 3. Analyzing the phenomenon of confidence collapse by gradient descent with finite step size is an interesting open problem. Moreover, our current analysis is for binary classification, and it is interesting to extend it to multi-class problems.

In this work, we prove a separation law of membership privacy between a linear model and a two-layer network under confidence-based attacks: two-layer ReLU networks trained by gradient descent provably leak membership privacy, while linear models leak less membership privacy with the same training dataset and algorithm. Experiments show that confidence collapse occurs in the two-layer ReLU network but not in the one-layer counterpart. From the confidence scores, it is easy to infer whether a given sample is involved in the training phase. Our results shed light on understanding the membership privacy in neural networks.

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

## A    Proof of Theorem 1

*Proof.* The weight of the first layer of the neural network is $(\boldsymbol{w}_1, ..., \boldsymbol{w}_k) \in \mathbb{R}^{d \times k}$, the weight of the second layer is $(v_1, ..., v_k)$. We denote $J := [k]$, $J_+ = \{i \in [k] | v_i \geq 0\}$, $J_- = \{i \in [k] | v_i < 0\}$ and $I := [n]$, $I_+ = \{i \in [n] | y_i = 1\}$, $I_- = \{i \in [N] | y_i = 1\}$. Notice that $I_+$ and $I_-$ are non-empty as the dataset contains at least one sample with label 1 or -1, and thus $J_+$ and $J_-$ are also non-empty. Denote $\mathbf{X} = \{\boldsymbol{x}_i\}$, $p := \max_{i,j \in I, x_j \neq \pm x_i} |\boldsymbol{x}_i^T \boldsymbol{x}_j|$, and $q := \min_{i \in [n]} ||\boldsymbol{x}_i||_2^2$. In this work we mainly use the following lemma from Lyu & Li (2019).

**Lemma 1** (Theorem 4.4 of (Lyu & Li, 2019)). *Let $\Phi(\cdot; \boldsymbol{\theta})$ be a homogeneous neural network parameterized by $\boldsymbol{\theta}$. Consider minimizing either the exponential or the logistic loss over a binary classification dataset $\{(\boldsymbol{x}_i, y_i)\}_{i=1}^n$ using gradient flow. Assume that there exists time $t_0$ such that $L(\theta(t_0)) < 1$, namely, $y_i \Phi(\boldsymbol{x}_i; \boldsymbol{\theta}(t_0);) > 0$ for every $x_i$. Then, gradient flow converges in direction to a first order stationary point (KKT point) of the following maximum margin problem in parameter space:*

$$\min_{\boldsymbol{\theta}} \frac{1}{2} ||\boldsymbol{\theta}||_2^2 \quad s.t. \quad y_i \Phi(\boldsymbol{x}_i; \boldsymbol{\theta}) \geq 1, \quad \forall i \in [n]. \tag{7}$$

With Lemma 1, gradient flow on the two-layer ReLU network $\Phi$ with weight $\mathbf{W}$ and $\mathbf{v}$ converges in direction to the KKT point of a maximum margin problem

$$\min_{\boldsymbol{\theta}} \frac{1}{2} \left( ||\mathbf{v}||_2^2 + \sum_{i=1}^k ||\boldsymbol{w}_i||_2^2 \right), \quad s.t. \quad y_i \Phi(\boldsymbol{x}_i) = y_i \sum_{i=1}^k v_i \sigma(\boldsymbol{w}_i^T \boldsymbol{x}_i) \geq 1, \quad \forall i \in [n],$$

where $k$ is the width of the network.

We are now ready to prove Theorem 1 by contradiction. Assume there exists $l \in [n]$ such that $y_l \Phi(\boldsymbol{x}_l) > 1$. Then by Corollary 2, $\lambda_l = 0$. If $l \in I_+$, then $y_l = 1$ and

$$1 < y_l \Phi(\boldsymbol{x}_l) = \sum_{i=1}^k v_i \sigma(\boldsymbol{w}_i^T \boldsymbol{x}_l)$$
$$\leq \sum_{j \in J_+} v_j \sigma(\boldsymbol{w}_j^T \boldsymbol{x}_l).$$

If $x_{l_-} = -x_l \notin X$, by Eq. 5 we have

$$
\begin{aligned}
\sum_{j \in J_+} v_j \sigma(\boldsymbol{w}_j^T \boldsymbol{x}_l) &= \sum_{j \in J_+} v_j \sigma(\sum_{i \in I} \lambda_i y_i v_j \sigma'_{i,j} \boldsymbol{x}_i^T \boldsymbol{x}_l) \\
&\leq \sum_{j \in J_+} v_j \sigma(\sum_{i \in I/\{l\}} \lambda_i y_i v_j \sigma'_{i,j} \boldsymbol{x}_i^T \boldsymbol{x}_l) \\
&\leq \sum_{j \in J_+} v_j \sum_{i \in I/\{l\}} \lambda_i |y_i v_j| \sigma'_{i,j} p \\
&\leq \sum_{j \in J_+} \sum_{i \in I} \lambda_i v_j^2 \sigma'_{i,j} p.
\end{aligned}
$$

If $x_{l_-} = -x_l \in X$, by Eq. 5 we have

$$
\begin{aligned}
\sum_{j \in J_+} v_j \sigma(\boldsymbol{w}_j^T \boldsymbol{x}_l) &= \sum_{j \in J_+} v_j \sigma(\sum_{i \in I} \lambda_i y_i v_j \sigma'_{i,j} \boldsymbol{x}_i^T \boldsymbol{x}_l) \\
&\leq \sum_{j \in J_+} v_j \sigma(\sum_{i \in I/\{l,l_-\}} \lambda_i y_i v_j \sigma'_{i,j} \boldsymbol{x}_i^T \boldsymbol{x}_l + \lambda_{l_-} y_{l_-} v_j \sigma'_{l_-,j} \boldsymbol{x}_{l_-}^T \boldsymbol{x}_l) \\
&\leq \sum_{j \in J_+} \sum_{i \in I} \lambda_i v_j^2 \sigma'_{i,j} p.
\end{aligned}
$$

So $\sum_{j \in J_+} v_j \sigma(\boldsymbol{w}_j^T \boldsymbol{x}_l) \leq \sum_{j \in J_+} \sum_{i \in I} \lambda_i v_j^2 \sigma'_{i,j} p$ always holds, then we have

$$
\begin{aligned}
\sum_{j \in J_+} v_j \sigma(\boldsymbol{w}_j^T \boldsymbol{x}_l) &\leq \sum_{j \in J_+} \sum_{i \in I} \lambda_i v_j^2 \sigma'_{i,j} p \\
&\leq p|I| \max_{i \in I} \sum_{j \in J_+} \lambda_i v_j^2 \sigma'_{i,j} \\
&= np \max_{i \in I} \sum_{j \in J_+} \lambda_i v_j^2 \sigma'_{i,j}.
\end{aligned}
$$

Thus we have $\max_{i \in I} \sum_{j \in J_+} \lambda_i v_j^2 \sigma'_{i,j} > \frac{1}{np}$. Analogously we have $\max_{i \in I} \sum_{j \in J_-} \lambda_i v_j^2 \sigma'_{i,j} > \frac{1}{np}$ if $l \in I_-$. Because the symmetric of $J_+$ and $J_-$ we can assume w.l.o.g. that $\max_{i \in I} \sum_{j \in J_+} \lambda_i v_j^2 \sigma'_{i,j} \geq \max_{i \in I} \sum_{j \in J_-} \lambda_i v_j^2 \sigma'_{i,j}$. Consider $r = arg \max_{i \in I} \sum_{j \in J_+} \lambda_i v_j^2 \sigma'_{i,j}$, we must have $\lambda_r > 0$ as $\sum_{j \in J_+} \lambda_r v_j^2 \sigma'_{r,j} > \frac{1}{np} > 0$. Thus by Corollary 2, $y_r \Phi(x_r) = 1$.

Denote $x_{r_-} = -x_r$, By Eq. 5, we have

$$
\begin{aligned}
\boldsymbol{w}_j^T \boldsymbol{x}_r &= (\sum_{i \in I} \lambda_i y_i v_j \sigma'_{i,j} \boldsymbol{x}_i^T) \boldsymbol{x}_r \\
&= \sum_{i \in I} \lambda_i y_i v_j \sigma'_{i,j} \boldsymbol{x}_i^T \boldsymbol{x}_r \\
&= y_r v_j (\lambda_r \sigma'_{r,j} - \mathbb{I}(\{\boldsymbol{x}_{r_-} \in \mathbf{X}\}) \lambda_{r_-} \sigma'_{r_-,j}) ||\boldsymbol{x}_r||_2 + \sum_{i \in I/\{r,r_-\}} \lambda_i y_i v_j \sigma'_{i,j} \boldsymbol{x}_i^T \boldsymbol{x}_r.
\end{aligned}
\tag{8}
$$

Now we consider two case: $r \in I_-$ and $r \in I_+$.

**Case 1.** $r \in I_-$, $\forall j \in J_+$. If $\boldsymbol{w}_j^T \boldsymbol{x}_r \geq 0$, then $\sigma'_{r,j} \geq 0$, $\boldsymbol{w}_j^T \boldsymbol{x}_{r_-} \leq 0$, $\sigma'_{r_-,j} = 0$

$$
\begin{aligned}
0 &\leq \boldsymbol{w}_j^T \boldsymbol{x}_r \\
&= y_r v_j (\lambda_r \sigma'_{r,j} - \mathbb{I}(\{\boldsymbol{x}_{r_-} \in \mathbf{X}\}) \lambda_{r_-} \sigma'_{r_-,j}) ||\boldsymbol{x}_r||_2 + \sum_{i \in I/\{r,r_-\}} \lambda_i y_i v_j \sigma'_{i,j} \boldsymbol{x}_i^T \boldsymbol{x}_r \\
&= -v_j \lambda_r \sigma'_{r,j} ||\boldsymbol{x}_r||_2 + \sum_{i \in I/\{r,r_-\}} \lambda_i y_i v_j \sigma'_{i,j} \boldsymbol{x}_i^T \boldsymbol{x}_r \\
&\leq -\lambda_r v_j \sigma'_{r,j} q + \sum_{i \in I/\{r,r_-\}} \lambda_i v_j \sigma'_{i,j} p.
\end{aligned}
$$

So $\lambda_r v_j \sigma'_{r,j} q \leq \sum_{i \in I/\{r,r_-\}} \lambda_i v_j \sigma'_{i,j} p$ when $\boldsymbol{w}_j^T \boldsymbol{x}_r \geq 0$.

If $\boldsymbol{w}_j^T \boldsymbol{x}_r < 0$, then $\sigma'_{r,j} = 0$ and

$$
\sum_{i \in I/\{r\}} \lambda_i v_j \sigma'_{i,j} p \geq 0 = \lambda_r v_j \sigma'_{r,j} q.
$$

Thus $\lambda_r v_j \sigma'_{r,j} q \leq \sum_{i \in I/\{r,r_-\}} \lambda_i v_j \sigma'_{i,j} p$ holds for all $j \in J_+$. So we have

$$
\begin{aligned}
\sum_{j \in J_+} \lambda_r v_j^2 \sigma'_{r,j} &\leq \frac{p}{q} \sum_{j \in J_+} v_j \sum_{i \in I/\{r,r_-\}} \lambda_i v_j \sigma'_{i,j} \\
&\leq \frac{np}{q} \max_{i \in I} \sum_{j \in J_+} \lambda_i v_j^2 \sigma'_{i,j} \\
&\leq \frac{1}{3} \max_{i \in I} \sum_{j \in J_+} \lambda_i v_j^2 \sigma'_{i,j}.
\end{aligned}
$$

This is in contradiction to the choice of $r$.

**Case 2.** $r \in I_+$

$$
\begin{aligned}
1 = y_r \Phi(x_r) &= \sum_{i=1}^{k} v_i \sigma(\boldsymbol{w}_i^T \boldsymbol{x}_r) \\
&= \sum_{j \in J_+} v_j \sigma(\boldsymbol{w}_j^T \boldsymbol{x}_r) + \sum_{j \in J_-} v_j \sigma(\boldsymbol{w}_j^T \boldsymbol{x}_r) \\
&= \sum_{j \in J_+} v_j (\boldsymbol{w}_j^T \boldsymbol{x}_r) \sigma'_{r,j} + \sum_{j \in J_-} v_j \sigma(\boldsymbol{w}_j^T \boldsymbol{x}_r).
\end{aligned}
$$

Consider $\sum\limits_{j \in J_+} v_i(\boldsymbol{w}_j^T \boldsymbol{x}_r)\sigma'_{r,j}$. Notice that $\sigma'_{r,j} > 0$ induces $\sigma'_{r_-,j} = 0$. By Eq. 8, we have

$$\sum_{j \in J_+} v_j(\boldsymbol{w}_j^T \boldsymbol{x}_r)\sigma'_{r,j}$$

$$= \sum_{j \in J_+} v_j \sigma'_{r,j} \left( y_r v_j (\lambda_r \sigma'_{r,j} - \mathbb{I}(\{\boldsymbol{x}_{r_-} \in \mathbf{X}\})\lambda_{r_-}\sigma'_{r_-,j})\|\boldsymbol{x}_r\|_2 + \sum_{i \in I/\{r,r_-\}} \lambda_i y_i v_j \sigma'_{i,j}\boldsymbol{x}_i^T \boldsymbol{x}_r \right)$$

$$= \sum_{j \in J_+, \sigma'_{r,j} > 0} \left( \lambda_r v_j^2 \sigma'_{r,j}\|\boldsymbol{x}_r\|_2 + \sum_{i \in I/\{r,r_-\}} \lambda_i y_i v_j^2 \sigma'_{i,j}\boldsymbol{x}_i^T \boldsymbol{x}_r \right)$$

$$\geq \sum_{j \in J_+, \sigma'_{r,j} > 0} \left( \lambda_r v_j^2 \sigma'_{r,j} q - \sum_{i \in I/\{r,r_-\}} \lambda_i v_j^2 \sigma'_{i,j} p \right)$$

$$\geq q \sum_{j \in J_+, \sigma'_{r,j} > 0} \lambda_r v_j^2 \sigma'_{r,j} + q \sum_{j \in J_+, \sigma'_{r,j} = 0} \lambda_r v_j^2 \sigma'_{r,j} - np \max_{i \in I} \sum_{j \in J_+} \lambda_i v_j^2 \sigma'_{i,j}$$

$$= (q - np) \max_{i \in I} \sum_{j \in J_+} \lambda_i v_j^2 \sigma'_{i,j}.$$

Consider $\sum\limits_{j \in J_-} v_j \sigma(\boldsymbol{w}_j^T \boldsymbol{x}_r)$, by Eq. 8 we have

$$\sum_{j \in J_-} v_j \sigma(\boldsymbol{w}_j^T \boldsymbol{x}_r)$$

$$= \sum_{j \in J_-, \sigma'_{r,j} > 0} v_j \left( y_r v_j (\lambda_r \sigma'_{r,j} - \mathbb{I}(\{\boldsymbol{x}_{r_-} \in \mathbf{X}\})\lambda_{r_-}\sigma'_{r_-,j})\|\boldsymbol{x}_r\|_2 + \sum_{i \in I/\{r,r_-\}} \lambda_i y_i v_j \sigma'_{i,j}\boldsymbol{x}_i^T \boldsymbol{x}_r \right)$$

$$= \sum_{j \in J_-, \sigma'_{r,j} > 0} v_j \left( v_j \lambda_r \sigma'_{r,j}\|\boldsymbol{x}_r\|_2 + \sum_{i \in I/\{r,r_-\}} \lambda_i y_i v_j \sigma'_{i,j}\boldsymbol{x}_i^T \boldsymbol{x}_r \right)$$

$$\geq \sum_{j \in J_-, \sigma'_{r,j} > 0} \left( v_j^2 \lambda_r \sigma'_{r,j} q - \sum_{i \in I/\{r,r_-\}} \lambda_i v_j^2 \sigma'_{i,j} p \right)$$

$$\geq \sum_{j \in J_-} v_j^2 \lambda_r \sigma'_{r,j} q - \sum_{j \in J_-} \sum_{i \in I/\{r,r_-\}} \lambda_i v_j^2 \sigma'_{i,j} p$$

$$\geq q \sum_{j \in J_-} v_j^2 \lambda_r \sigma'_{r,j} - np \max_{i \in I} \sum_{j \in J_-} \lambda_i v_j^2 \sigma'_{i,j}.$$

Combining the above results we have

$$1 \geq \sum_{j \in J_+} v_i(\boldsymbol{w}_j^T \boldsymbol{x}_r)\sigma'_{r,j} + \sum_{j \in J_-} v_j \sigma(\boldsymbol{w}_j^T \boldsymbol{x}_r)$$

$$\geq (q - np) \max_{i \in I} \sum_{j \in J_+} \lambda_i v_j^2 \sigma'_{i,j} - np \max_{i \in I} \sum_{j \in J_-} \lambda_i v_j^2 \sigma'_{i,j}$$

$$\geq (q - 2np) \max_{i \in I} \sum_{j \in J_+} \lambda_i v_j^2 \sigma'_{i,j}$$

$$> (q - 2np)\frac{1}{np} > 1.$$

Thus, we reach a contradiction. □

## B    Proof of Proposition 1

Consider the new training set $\{(\boldsymbol{x}_i' := [\boldsymbol{x}_i, \mathbf{a}_i], y_i)\}_{i=1}^n$. Firstly, the norm of the new sample $\boldsymbol{x}_i'$ is given by $\sqrt{||\boldsymbol{x}_i||_2^2 + ||\mathbf{a}_i||_2^2} = \sqrt{||\boldsymbol{x}_i||_2^2 + 3np'}$. The inner product of $\boldsymbol{x}_i'$ and $\boldsymbol{x}_j'$ ($i \neq j$) is $\boldsymbol{x}_i'^T \boldsymbol{x}_j' = \boldsymbol{x}_i^T \boldsymbol{x}_j + \mathbf{a}_i^T \mathbf{a}_j = \boldsymbol{x}_i^T \boldsymbol{x}_j$. Thus

$$p := \max_{i,j \in [n], \boldsymbol{x}_j \neq \pm \boldsymbol{x}_i} |\boldsymbol{x}_i'^T \boldsymbol{x}_j'| \leq \max_{i,j \in I} |\boldsymbol{x}_i^T \boldsymbol{x}_j| =: p'.$$

As the new dataset $\{(\boldsymbol{x}_i', y_i)\}_{i=1}^n$ is in $\mathbb{R}^{n+d} \times \{-1, 1\}$, we have a) $n + d \geq n$, and b) $\frac{\min_{i \in [n]} ||\boldsymbol{x}_i'||_2^2}{3p} \geq \frac{3np'}{3p} \geq \frac{3np}{3p} = n$. Thus $n \leq \min\{n + d, \frac{\min_{i \in [n]} ||\boldsymbol{x}_i'||_2^2}{3p'}\}$ always holds. By applying Theorem 1 to the new dataset $\{(\boldsymbol{x}_i', y_i)\}_{i=1}^n$, we have that the confidence collapse occurs on $\{(\boldsymbol{x}_i', y_i)\}_{i=1}^n$.

## C    Proof of Theorem 2

*Proof.* We calculate the probabilistic mass of $\{\boldsymbol{x} \in \mathbb{R}^d : \Phi(\boldsymbol{x}) = \pm\gamma\}$ under the distribution $\mathcal{D}$. Firstly, we consider the solution of $\Phi(\boldsymbol{x}) = \pm\gamma$. By the stationarity condition of the KKT point (Corollary 1), we have $\boldsymbol{w}_s = \sum_{i \in I} \lambda_i y_i v_s \sigma_{i,s}' \boldsymbol{x}_i$. Plugging it into $\Phi(\boldsymbol{x}) = \pm\gamma$, where $\boldsymbol{x} = (x^{(1)}, ..., x^{(d)})$, we obtain

$$\pm\gamma = \Phi(\boldsymbol{x}) = \sum_{j \in J} v_j \sigma(\boldsymbol{w}_j^T \boldsymbol{x}) = \sum_{j \in J} v_j \mathbb{I}(\{\boldsymbol{w}_j^T \boldsymbol{x} > 0\}) \boldsymbol{w}_j^T \boldsymbol{x}$$

$$= \sum_{j \in J} v_j \mathbb{I}(\{\boldsymbol{w}_j^T \boldsymbol{x} > 0\}) \sum_{i=1}^d w_{ji} x_i = \sum_{i=1}^d (\sum_{j \in J} v_j \mathbb{I}(\{\boldsymbol{w}_j^T \boldsymbol{x} > 0\}) w_{ji}) x_i.$$

Denote $a_j(\boldsymbol{x}) = \sum_{j \in J} v_j \mathbb{I}(\{\boldsymbol{w}_j^T \boldsymbol{x} > 0\}) w_{ji}$, notice $a_j(\boldsymbol{x})$ can take at most $2^j$ values for different $\boldsymbol{x}$, thus $\mathbf{a}(\boldsymbol{x}) = (a_1(\boldsymbol{x}), ..., a_d(\boldsymbol{x}))$ has at most $2^{jd}$ different choices. Denote $A := \{\mathbf{a}(\boldsymbol{x}) | \boldsymbol{x} \in \mathbb{R}^d\}$, $S(\mathbf{a}) := \{\boldsymbol{x} \in \mathbb{R}^d | \mathbf{a}^T \boldsymbol{x} = \pm\gamma\}$ we must have $R(\mathbf{X}, \Phi) \subset \cup_{\mathbf{a} \in A} S(\mathbf{a})$. Notice $S(\mathbf{a})$ is just two hyperplanes $\mathbf{a}^T \boldsymbol{x} = \pm\gamma$, for an arbitrary continuous distribution on $\mathbb{R}^d$, assuming w.l.o.g. $\mathbf{a}^T = (||\mathbf{a}||_2, 0, ..., 0)$ and considering the case $\mathbf{a}^T \boldsymbol{x} = 1$, we have

$$\Pr(S(\mathbf{a}), \mathbf{a}^T \boldsymbol{x} = \gamma) = \Pr(\{\boldsymbol{x} | x_1 = \frac{\gamma}{||\mathbf{a}||_2}\})$$

$$= \lim_{\delta \to 0} \left( \Pr(\{\boldsymbol{x} | x_1 \leq \frac{\gamma}{||\mathbf{a}||_2}\}) - \Pr(\{\boldsymbol{x} | x_1 \leq \frac{\gamma}{||\mathbf{a}||_2} - \delta\}) \right)$$

$$= 0.$$

Analogously $\Pr(S(\mathbf{a}), \mathbf{a}^T \boldsymbol{x} = -\gamma) = 0$, so we have $\Pr(S(\mathbf{a})) = 0$. Thus

$$\Pr(R(\mathbf{X}, \Phi)) \leq \sum_{\mathbf{a} \in A} S(\mathbf{a}) = |A| \cdot 0 = 0.$$

So for an arbitrary $\boldsymbol{x}_{test}$ sampled from $\mathcal{D}$, $\Pr_{\mathcal{D}}(\boldsymbol{x}_{test} \in R(\mathbf{X}, \Phi)) = 0$ □

## D    Proof of Theorem 3

*Proof.* We prove Theorem 3 by constructing a hard instance, which satisfies the assumption on the training dataset but does not suffer from confidence collapse by one-layer neural networks. Based on Eq. 3, the max-margin problem is formulated by

$$\min_{\boldsymbol{\theta}} \frac{1}{2}(||\boldsymbol{w}||_2^2 + b^2), \quad \text{s.t.} \quad y_i(\boldsymbol{w}^T \boldsymbol{x}_i + b) \geq 1, \quad \forall i \in [n].$$

Consider $(\boldsymbol{x}_1, +1), (\boldsymbol{x}_2, +1), (\boldsymbol{x}_3, +1), (\boldsymbol{x}_4, -1)$, where $\boldsymbol{x}_i \in \mathbb{R}^d, d > 4, \boldsymbol{x}_2 = -\boldsymbol{x}_1, \boldsymbol{x}_3^T \boldsymbol{x}_1 = 0, \boldsymbol{x}_4^T \boldsymbol{x}_1 = 0, \boldsymbol{x}_3^T \boldsymbol{x}_4 = -\epsilon$, and $\epsilon$ is an arbitrary positive constant less than $1/3$ (e.g., $\boldsymbol{x}_1 = (\sqrt{d}, 0, ..., 0)$, $\boldsymbol{x}_2 = (-\sqrt{d}, 0, ..., 0)$, $\boldsymbol{x}_3 = (0, \sqrt{d}, 0, ..., 0)$, and $\boldsymbol{x}_4 = (0, \frac{-\epsilon}{\sqrt{d}}, \sqrt{d - \frac{\epsilon^2}{d}}, 0, ..., 0)$).

Firstly, by $q := \min_{i \in [n]} ||\boldsymbol{x}_i||_2 = d$ and $p := \max_{i,j \in [n], \boldsymbol{x}_j \neq \pm \boldsymbol{x}_i} |\boldsymbol{x}_i^T \boldsymbol{x}_j| = \epsilon$, we can easily verify that $4 \leq \min\{d, \frac{q}{3p}\}$. Besides, we can use a one-layer neural network $\Phi(\boldsymbol{x}) = (\boldsymbol{x}_3 - \boldsymbol{x}_4)^T \boldsymbol{x} + 1$ to classify them correctly:

$$\boldsymbol{w}^T \boldsymbol{x}_1 + b = 1 > 0,$$
$$\boldsymbol{w}^T \boldsymbol{x}_2 + b = 1 > 0,$$
$$\boldsymbol{w}^T \boldsymbol{x}_3 + b = d + |\epsilon| + 1 > 0,$$
$$\boldsymbol{w}^T \boldsymbol{x}_4 + b = -d - |\epsilon| + 1 < 0,$$

which means the dataset is linearly separable. Next, following the KKT stationarity condition of the optimization problem Eq. 6, there exist $\lambda_i \geq 0, i \in [4]$ such that $\boldsymbol{w} = \sum_{i=1}^4 \lambda_i y_i \boldsymbol{x}_i$. For the optimization problem Eq. 6, confidence collapse only occurs with $y_i \Phi(\boldsymbol{x}_i) = 1, i \in [4]$. Otherwise by complementary slackness we will have $\lambda_i = 0, i \in [4]$, which implies $\boldsymbol{w} = b = 0$ and all samples are misclassified. Pluging $\boldsymbol{w}$ into $y_i \Phi(\boldsymbol{x}_i) = 1, i \in [4]$, we have

$$\boldsymbol{w}^T \boldsymbol{x}_1 + b = \sum_{i=1}^4 \lambda_i y_i \boldsymbol{x}_i^T \boldsymbol{x}_1 + b = (\lambda_1 - \lambda_2)d + b = 1,$$

$$\boldsymbol{w}^T \boldsymbol{x}_2 + b = \sum_{i=1}^4 \lambda_i y_i \boldsymbol{x}_i^T \boldsymbol{x}_2 + b = (\lambda_2 - \lambda_1)d + b = 1,$$

which induce $b = 1$. By

$$\boldsymbol{w}^T \boldsymbol{x}_3 + b = \sum_{i=1}^4 \lambda_i y_i \boldsymbol{x}_i^T \boldsymbol{x}_3 + b = \lambda_3 d + \lambda_4 |\epsilon| + 1 = 1,$$

we have $\lambda_3 = \lambda_4 = 0$, at this time

$$\boldsymbol{w}^T \boldsymbol{x}_4 + b = \sum_{i=1}^4 \lambda_i y_i \boldsymbol{x}_i^T \boldsymbol{x}_4 + b = -\lambda_3 |\epsilon| - \lambda_4 d + 1 = 1,$$

which reaches a contradiction ($y_4 = -1$).

Denote by $\gamma = \arg\min_{\boldsymbol{x} \in \mathbf{X}} |\boldsymbol{w}^T \boldsymbol{x} + b|$ and $\Gamma = \arg\max_{\boldsymbol{x} \in \mathbf{X}} |\boldsymbol{w}^T \boldsymbol{x} + b|$. Confidence non-collapse implies $\Gamma$ is strictly larger than $\gamma$. By Definition 1, we have $R(\mathbf{X}, \Phi) = \{\boldsymbol{x} \in \mathbb{R}^d : \gamma \leq |\boldsymbol{w}^T \boldsymbol{x} + b| \leq \Gamma\}$. We can rotate the coordinate such that $\boldsymbol{w}^T = (||\boldsymbol{w}||_2, 0, ..., 0)$. Let $\boldsymbol{x} = (x^{(1)}, ..., x^{(d)})$ and $\mathcal{D}'$ be the new distribution after rotation. We have

$$\Pr_{\mathcal{D}}(R(\mathbf{X}, \Phi)) = \Pr_{\mathcal{D}'} \left( \left\{ \boldsymbol{x} : \frac{\gamma - b}{||\boldsymbol{w}||_2} \leq x^{(1)} \leq \frac{\Gamma - b}{||\boldsymbol{w}||_2} \right\} \cup \left\{ \boldsymbol{x} : \frac{-\Gamma - b}{||\boldsymbol{w}||_2} \leq x^{(1)} \leq \frac{-\gamma - b}{||\boldsymbol{w}||_2} \right\} \right).$$

As $\Pr_{\mathcal{D}'} \left( \{ \boldsymbol{x} : \frac{\gamma - b}{||\boldsymbol{w}||_2} \leq x^{(1)} \leq \frac{\Gamma - b}{||\boldsymbol{w}||_2} \} \right) = F_{\mathcal{D}'}^{(1)}(\frac{\Gamma - b}{||\boldsymbol{w}||_2}) - F_{\mathcal{D}'}^{(1)}(\frac{\gamma - b}{||\boldsymbol{w}||_2}) > 0$ (this is because $\mathcal{D}$ is supported on $\mathbb{R}^d$), where $F_{\mathcal{D}'}^{(i)}$ is the marginal cumulative density function of $\mathcal{D}'$ on axis $i$, we have $\Pr_{\mathcal{D}}(R(\mathbf{X}, \Phi)) > 0$. $\square$

