# OpenReview forum: "A Separation Law of Membership Privacy between One- and Two-Layer Networks "
_TMLR — Withdrawn by Authors_

### Review · Reviewer_krB7 · 2022-12-29

**Summary Of Contributions:**

The paper considers the "confidence collapse" phenomenon, where neural networks converge over the course of training to solutions with concentrated confidence scores. They prove that confidence collapse happens for certain overparameterized networks, and that it does not happen for single layer networks. Because confidence collapse implies high performance membership inference attacks, they are able to show that overparameterized networks are vulnerable to attack.

**Audience:**

Yes

**Broader Impact Concerns:**

No concerns here.

**Claims And Evidence:**

Yes

**Requested Changes:**

I would like more details about the experiments. Does training use cross entropy loss? The confidence reported in the experiments is not a standard definition, so I would like to know what is being reported. I worry that the confidence collapse they observe in practice is simply training to 0 loss, especially since they are training for 1000s of epochs.

Please clarify whether the theoretical analysis implies that the network tends towards 0 loss. If this is the case, it significantly limits the interestingness of the result to the privacy community.

**Strengths And Weaknesses:**

Strengths:

As far as I can tell, the confidence collapse phenomenon the paper proposes is new, and bears some relation to the neural collapse phenomenon discussed in prior work

The experiments seem to support the claims of the paper.

Weaknesses:

It seems that confidence collapse may be implied by the network converging to a 0 training loss solution. In the experiments, they run training for 1000s of epochs, at which point I'm maybe unsurprised they are converging to a solution with very low training loss.

How is the absolute value of confidence 20? Shouldn't it be bounded between 0 and 1? I would like to know more about the experimental setup.

The data transformation proposed in 4.1.1 does not reflect any practical setting.

Recent work in membership inference suggests that every example has its own "hardness", and calibrating attacks per-example (as in the first reference in the paper) significantly improves performance over the simple loss thresholding proposed in prior work and used as a basis for this work. This makes me worry that the insights from this paper do not generalize to real-world settings.

---

> ### Author Response · Authors · 2023-02-27
> **Authors Response to Reviewer krB7**
>
> *Q1. It seems that confidence collapse may be implied by the network converging to a 0 training loss solution.*
>
> A1.  Our theoretical analysis does not imply that the network converges to 0 loss. The main theory (Theorem 1) of our paper is based on Theorem 4.4 of (Lyu & Li, 2019), which shows that the gradient flow of a homogeneous neural network converges to a KKT point of a maximum margin problem. This KKT point does not directly imply zero loss.
>
> *Q2. How is the absolute value of confidence 20? Shouldn't it be bounded between 0 and 1? I would like to know more about the experimental setup.*
>
> A2. In the experiments, we train model with cross-entropy loss and report the logits as the confidence score. Thus, the confidence score report in our experiments are not necessary between 0 and 1. One can normalize the logit confidence to [0,1] by applying the Sigmoid activation.
>
>
> *Q3. The data transformation proposed in 4.1.1 does not reflect any practical setting.*
>
> A3. One of the scenarios that our data transformation proposed in 4.1.1 can appear is padding the training data with its identity pseudo-label. The pseudo-labels provides additional information about the identity of each sample and thus will not conflict with the 100\% accuracy assumption of the classifier. For example, when training a recommendation network with products and users, we may use the unique ID of products and users as additional features to improve the performance. We agree this data transformation has its own limitation and we hope it can provide further insight about resolving the small data assumption.
>
> *Q4. The insights from this paper do not generalize to real-world settings.*
>
> A4. Our work theoreticallty shows that two-layer ReLU networks trained by gradient descent **provably** leak membership privacy under confidence-based attacks. If other MIA attacks e.g. calibrating attacks are significantly better than the confidence-based attacks, then two-layer ReLU networks should also **provably** leak membership privacy under calibrating attacks, otherwise we cannot conclude that calibrating attacks per-example are better than the confidence-based attacks in every scenario. Thus, we believe our work is interesting for the MIA community and can inspire more theoretical analysis in this area.

---

### Review · Reviewer_kXVZ · 2023-01-02

**Summary Of Contributions:**

This paper provides a theoretical analysis of the membership inference vulnerability of single-layer and two-layer ReLU networks. To this end, the paper proposes the privacy-preservation region. The paper shows that the area is ~0 in the two-layer network, which is the confidence collapse phenomenon observed in the prior work. In evaluation with the binarized MNIST and CIFAR-10 tasks, the paper shows that the privacy-preserving region is almost zero over multiple runs.

**Audience:**

Yes

**Claims And Evidence:**

No

**Requested Changes:**

1. Clarification of the definition of the privacy-preserving region.
2. Clarification of the connection between membership privacy and the region above.
3. Clarification of the connection between the membership inference and the confidence collapse.
4. Extension of the theoretical analysis on a binary classifier to multi-class classifiers.
5. Evaluation with the membership inference attacks on multi-class classifiers.

**Strengths And Weaknesses:**

[Strengths]

1. The paper connects membership inference vulnerability with the confidence collapse phenomenon.
2. The paper conducts a theoretical analysis between the membership inference success and single-/two-layer ReLU networks.
3. The paper shows that two-layer ReLU networks, trained for a binarized task, are more vulnerable than single-layer ReLU networks.

[Weaknesses]

1. It's a bit unclear why the distance between the max. and min. confidence values is the "privacy-preserving region."
2. Related to 1., it's also unclear the connection between the membership inference success and confidence collapse.
3. The paper only studies binary classification problems; it's unclear we can extend this to practical attacks on multi-class models.
4. The evaluation does not perform any membership inference attacks, so it's unclear whether the one-layer networks actually preserve privacy more.

[Detailed comments]

Unclear connection between the confidence values and the privacy-preserving region.
> It's hard to accept that if a ~= b, then the membership inference attack's success will be high. Suppose that 1) a ~= b == $\tau$ and 2) the attacker has a set of samples (containing 50% of members and 50% of non-members). How does the attacker know the $\tau$ in advance? If the attacker does not know the $\tau$ then the attacker necessarily has false-positives in their inferences. Therefore, I cannot accept the definition of the privacy-preserving region.

> Instead, the paper can restate that there is an approximate way to quantify the membership inference vulnerability as the distance between the min and max confidence scores. If this is the route taken, the paper should compare it with other metrics, such as the attacker's advantage or the membership inference attack success @ 0.1% false-positive region, and show why the proposed definition is better or at least the same.

Unclear connection between the membership inference vulnerability and confidence collapse.
> Because of the issue discussed above, it's also become blurry why the confidence collapse makes the membership attackers succeed more. To make this connection explicit, the paper must show that the confidence collapse only happens for the training samples, and any test-time samples should be outside the confidence score. My intuition is that: if some of the test-time instances do have the same confidence score, then the attacker will end up with a lot of false-positive scores.

Potential limitation of the theoretical analysis.
> The paper only focuses on the vulnerability of binary classifiers against membership inference attacks. However, it overlooks that most prior work considers multi-class classification problems, which leads to much practical demonstration of the vulnerability. To have practical meanings, it has to be shown that we can straightforwardly extend the paper's analysis to multi-class classification problems.

Weak evaluation
> The entire paper discusses membership inference vulnerability, but the evaluation section only measures confidence collapse. The paper also conducts the experiments only with the binary classification tasks. It is hard to connect the results with prior work that conducts privacy attacks on multi-class classifiers, such as ResNets trained on CIFAR-10.

[Minor]
1. In Sec 1.1, what is the "threatened model?"

---

> ### Author Response · Authors · 2023-02-27
> **Authors Response to Reviewer kXVZ**
>
> Thanks for the reviewer's valuable comments.
>
> *Q1. Unclear connection between the confidence values and the privacy-preserving region.*
>
> A1. In our work, we assume that the attacker can access to the confidence score of the threatened model on one Trojan training example x_0 (See paragraph Threat model in 1.1). Thus, the attacker knows $\tau$ through access the Trojan example.
>
> *Q2. Unclear connection between the membership inference vulnerability and confidence collapse.*
>
> A2. In Theorem 2, we prove that the probability of test-time instances to have the same confidence score as the training examples is 0. Thus, the absolute value of the confidence score of a randomly test data almost surely differs from the confidence score of the training samples.
>
> *Q3. Potential limitation of the theoretical analysis.*
>
> A3. One potential way for extending our results to multi-class classification is to convert the problem to the binary case with one vs rest loss, i.e., we define the confidence score of multi-class classification as $f_y(x)-\max_{i\in [k],i\neq y}f_{i}(x)$ for a given sample (x,y) and classifier f. We will leave the detail analysis as a future work.
>
>
> *Q4. Weak evaluation*
>
> A4. Our work focuses on theoretically discussing the membership privacy in the one- and two-layer neural networks, and our experiments successfully validate our theoretical analysis. The confidence collapse theory on larger networks, e.g., ResNet is still missing, so we don’t conduct experiments on them. Based on the reviewer's suggestion, we run experiments with multi-class classification tasks on CIFAR10 with ResNet, and report the $f_y(x)-\max_{i\in [k],i\neq y}f_{i}(x)$ of the ground-truth labels as the confidence score of training images (See Fig. 2). We also identify the confidence collapse phenomenon, this is consist with the neural collapse we observed in deep neural networks.

---

### Review · Reviewer_L7bN · 2023-02-14

**Summary Of Contributions:**

The paper studies conditions under which ML models are susceptible to membership inference attacks (MIA).
They study a concept of confidence collapse and show that under certain conditions it can be used to show that MIA will always be possible. In particular they show that 2 layer networks are always subject to MIA (under some assumptions on the training data and training method) while on 1 layer networks there are examples on which the attacker will not be able to distinguish member vs non-members. The authors provide preliminary experiments of their hypothesis using MNIST and CIFAR datasets, as well as synthetic data.

**Audience:**

Yes

**Broader Impact Concerns:**



**Claims And Evidence:**

Yes

**Requested Changes:**

- Please consider explaining your result wrt recent work on evaluation of what MIA success ("Membership Inference Attacks From First Principles" by Carlini et al).
- Please explain how dataset size matters in practice --- what if one was duplicate the data
- 1 vs 2 layer model: would another way of stating it as linear model vs not be accurate or in terms of dataset, if it is linearly separable?
- differential privacy can protect against MIA; it would be interesting to see how this would fit into the analysis of this work wrt DP-SGD training of both types of networks that you consider
- please consider expanding on confidence collapse; is it something known in the literature, if not please consider giving examples
- some discussion on how your work inspires defences (e.g., would adding more data to not satisfy assumptions in Thm 1 be one?)
- Eq 4 in Thm 1 is not yet defined

**Strengths And Weaknesses:**

Strengths:
- understanding which models are more prone to MIA is an important topic
- the paper makes a nice step towards that, even if it is under certain conditions, the result is important

Weaknesses:
- connection between confidence collapse and MIA could be justified further
- discussion on differential privacy and how it fits this work
- mention of Trojan attacks in the intro is confusing and it is not clear if the method requires a malicious attacker

---

> ### Author Response · Authors · 2023-02-27
> **Authors Response to Reviewer L7bN**
>
> Thanks for the reviewer's valuable comments.
>
> *Q1: Please consider explaining your result wrt recent work on evaluation of what MIA success ("Membership Inference Attacks From First Principles" by Carlini et al).*
>
> A1: Carlini et al argued that attacks should be evaluated by computing their true-positive rate at low false-positive rates. In our work, we prove that confidence-based MIA on two-layer network can achieve 100% true-positive rate and 0% false-positive rate. Thus our results still have perfect performance under the metric proposed in Carlini et al.
>
> *Q2. Please explain how dataset size matters in practice.*
>
> A2. In practice, when the dataset size is large, confidence collapse will not occur on the two-layer network and we cannot use confidence-based attacks to achieve perfect MIA on two-layer networks. Duplication of images doesn’t affect confidence collapse in practice.
>
> *Q3. 1 vs 2 layer model: would another way of stating it as linear model vs not be accurate or in terms of dataset, if it is linearly separable?*
>
> A3. As our theoretical results are based on one or two layer models but not the dataset, we think our description is proper.
>
> *Q4. differential privacy can protect against MIA; it would be interesting to see how this would fit into the analysis of this work wrt DP-SGD training of both types of networks that you consider.*
>
> A4. In DP we add noise during training the network, it’s equal to increase the size of dataset and can definitely help to prevent two-layer networks from confidence collapse. Thus, with DP-SGD training, we are able to protect the model against MIA.
>
> *Q5. please consider expanding on confidence collapse; is it something known in the literature, if not please consider giving examples*
>
> A5. Confidence collapse is similar to the neural collapse, which we provide a detailed discussion in Sec. 2. The theoretical analysis of neural collapse depends on the layer-peeled assumption, which is too strong to fit the empirical situations. In our work, we theoretically prove that a weaker version of neural collapse - confidence collapse - occurs in gradient flow of two-layer neural network with only an assumption on the number of training samples.
>
>
> *Q6. some discussion on how your work inspires defences (e.g., would adding more data to not satisfy assumptions in Thm 1 be one?)*
>
> A6. You are right, adding more data is definitely a good way to defend the confidence-based attack proposed in our work. We will add more discussions in our revision.
>
> *Q7. mention of Trojan attacks in the intro is confusing and it is not clear if the method requires a malicious attacker.*
>
> A7. The Trojan example is required for applying confidence-based MIA in our settings, because the attackers need to know the confidence score of training examples.

---

### Note · Authors · 2023-05-17

I have read and agree with the venue's withdrawal policy on behalf of myself and my co-authors.